# Ultra-lightweight rechargeable battery with enhanced gravimetric energy densities >750 Wh kg$^{-1}$ in lithium–sulfur pouch cell
Kenji Kakiage [1] ✉, Toru Yano[1], Hiroki Uehara[2] & Masaki Kakiage [2] ✉

Lithium–sulfur (Li–S) rechargeable batteries have been expected to be lightweight energy storage devices with the highest gravimetric energy density at the single-cell level reaching up to 695 Wh kg$_{(cell)}^{-1}$, having also an ultralow rate of 0.005 C only in the first discharge. Sulfurized polyacrylonitrile (SPAN) is one of the sulfur-based active materials, which allows more freedom in the Li–S cell design because it shows no undesirable reactions with electrolyte solutions. Here we present an original Li–S pouch cell construction, ADEKA's Lithium–Sulfur/Pouch Cell (ALIS-PC). It is an ultra-lightweight rechargeable battery cell, which is designed by combining the SPAN cathode and effective ten technologies involving chemical engineering. As a result, the highest gravimetric energy densities of 713 and 761 Wh kg$_{(cell)}^{-1}$ after some charge-and-discharge cycles, which were based on the total mass of all cell components, were achieved with successful operating at 0.1 and 0.05C-rates, respectively, significantly exceeding those of commercial lithium-ion and next-generation rechargeable batteries in development.

Lithium–sulfur (Li–S) rechargeable batteries, which are composed of sulfur-based cathodes, liquid electrolytes, and lithium–metal anodes, have been actively investigated as extremely promising next-generation energy storage devices because of the low-temperature synthetic processes of cathode active materials, Co- and Ni-free systems, and the potential of high gravimetric energy density at the single-cell level (Wh kg$_{(cell)}^{-1}$). Much research has been done on improving Li–S battery performances, particularly with regard to chg./dischg. cycle lifespan. For example, excellent investigations on complexes of sulfur and carbons, adsorbents of eluted lithium polysulfides in electrolyte solutions, catalytic materials to promote redox reactions, electrolyte solutions with various additives or high salt concentrations, coated separators and lithium–metals for short-circuit or redox shuttle suppression, and others have been reported by scholarly experiments at coin or <1000 mAh cell levels[1–8]. On the other hand, experiments and verifications discussing the energy densities at higher capacities of >1 Ah cell levels are rather limited. In other words, there is insufficient integration of developed new materials and technologies, characteristic ideas and challenges, theoretical analyses, total cell designs, and cell-assembly techniques. This may be due to the lack of collaborations between academia and industry in the Li–S field. For Li–S single cells, energy densities of over 500 Wh kg$_{(cell)}^{-1}$ under

low-rate conditions have been reported to date in a small number of cases, with 695 Wh kg$_{(cell)}^{-1}$ for pouch type being the highest value, which, however, could be attained only under conditions of a first discharge at an impracticable ultralow rate of 0.005C[9]. The realization of ultra-lightweight rechargeable batteries with stable and realistic operations will in turn lead to the realization of energy-efficient electric vehicles (EVs) and mobile scooters, energy storage systems and battery stations, portable battery packs, drones, electric vertical take-off and landing (eVTOL) and high-altitude long-endurance (HALE) aircrafts, high-altitude platform stations (HAPSs), and others. The improved performance and commercialization of Li–S batteries will enrich our lives with the EV revolution and urban air mobility (UAM)[10–12].

Sulfurized polyacrylonitrile (SPAN) is one of the sulfur-based active materials receiving the most attention, and numerous papers on this material have been reported since 2002. The following features of SPAN have attracted the attention of researchers[13–20]. It has a Co- and Ni-free chemical structure consisting of ubiquitous elements, S–C covalent bonds, and polymeric backbones of cyclized PAN (Supplementary Fig. 1). It is amenable to simple process and low-temperature synthetic process under 500 °C, and has a moderate specific capacity of over 500 mAh g$^{-1}$

¹Environmental & Energy Materials Laboratory, ADEKA CORPORATION, Arakawa-ku, Tokyo, Japan. ²Division of Molecular Science, Graduate School of Science and Technology, Gunma University, Kiryu, Gunma, Japan. ✉e-mail: ke-kakiage@adeka.co.jp; kakiage@gunma-u.ac.jp

(Supplementary Fig. 2). It shows better power performance than conventional sulfur-based active materials, and various electrolytes from liquids to solids without polysulfide shuttling are available (Supplementary Fig. 3). It possesses superior adsorption abilities of lithium polysulfides (Supplementary Fig. 4). It has no µL-electrolyte/mg-sulfur (E/S) ratio restrictions

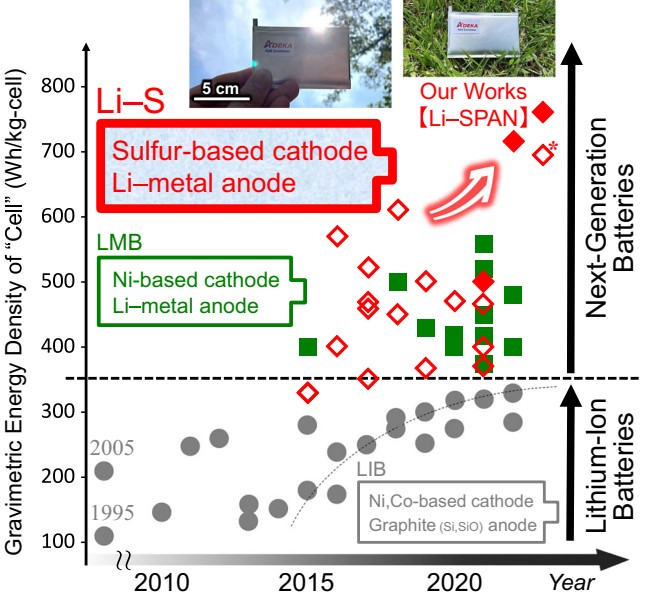

**Fig. 1 | Gravimetric energy densities of various rechargeable battery cells for our works, products, under developments, and researches.** The data (closed rhombuses) of the Li–SPAN battery cells, sulfurized polyacrylonitrile (SPAN) cathode | carbonate electrolyte solution | Li–metal anode, are for our works. The data (closed circles) of lithium-ion rechargeable batteries (LIBs) are for Panasonic Energy Co., Ltd., LG Chem/LG Energy Solution Ltd., Samsung SDI Co., Ltd., and Contemporary Amperex Technology Co., Limited. The data (open rhombuses and closed squares) of next-generation batteries [lithium–sulfur (Li–S) batteries and lithium–metal batteries (LMBs)] are for Enpower Japan Corp./Enpower Greentech Inc., SoftBank Corp., GS Yuasa Corporation, Sion Power Corporation, Cuberg, SES AI Corporation, and references[2,5,21–26]. An asterisk mark indicates an unstable chg./dischg. performance, the cell energy density is only in the first discharge[9]. Photographs show the Li–SPAN pouch cell with >750 Wh kg$_{(cell)}^{-1}$ fabricated in this work.

and it shows outstanding reversibility with almost 100% coulombic efficiency and over 500 cycles in charge-and-discharge (chg./dischg.) operations (Supplementary Fig. 5). Although these characteristics provide greater flexibility in Li–S cell design, the gravimetric energy densities of Li–SPAN cells have been limited to less than 500 Wh kg$_{(cell)}^{-1}$. In this work, an original Li–S pouch cell design, that is, ADEKA's Lithium–Sulfur/Pouch Cell (ALIS-PC), was realized by applying SPAN (ADEKA AMERANSA SAM-8) and ten state-of-the-art technologies. As a result, an 11 Ah ultra-lightweight Li–SPAN pouch cell with the energy densities of 713 and 761 Wh kg$_{(cell)}^{-1}$ at 0.1 and 0.05C-rates at 30 °C, respectively, after some chg./dischg. cycles was realized. These are the world record cell energy densities significantly exceeding those of commercial lithium-ion rechargeable batteries (LIBs) and new-type next-generation batteries under development such as lithium–metal batteries (LMBs) and lithium-air batteries (LABs) (Fig. 1 and Supplementary Table 1)[21–26].

## Results and Discussion
### Technology I
A 3D-Al foam sheet, Al-CELMET, was chosen as the current collector to increase the areal mass loading of sulfur in the SPAN cathode[27]. Figs. 2a and 2b respectively show a scanning electron microscopy (SEM) image of the SPAN cathode on the 3D-Al foam and the areal cathode capacities (mAh cm$^{-2}$) on the 3D-Al foam or conventional carbon-coated Al foil. A higher sulfur loading of 32.4 mg$_{(S)}$ cm$^{-2}$ (68.0 mg$_{(SPAN)}$ cm$^{-2}$) on both sides was achieved by stabilizing the thick SPAN layer in the 3D-Al foam, and four times the areal capacity of 46.6 mAh cm$^{-2}$ with a specific capacity of 686 mAh g$_{(SPAN)}^{-1}$ (1441 mAh g$_{(S)}^{-1}$) and 100% coulombic efficiency at 0.1C-rate (68 mA g$_{(SPAN)}^{-1}$) and 30 °C was obtained reversibly. A superior performance of 770 mAh g$_{(SPAN)}^{-1}$ was observed under the 0.01C-rate (6.8 mA g$_{(SPAN)}^{-1}$) condition.

### Technology II
The weight of 3D-Al foam with 96% porosity is 10 mg cm$^{-2}$; furthermore, a 31% weight saving was achieved by a laser-drilling technique with homogeneous holes of φ = 1.0 mm (Fig. 2c)[28]. On the other hand, further processing was not possible in terms of strength (Supplementary Fig. 6). Capacity losses due to the processing was not seen at lower chg./dischg. rates despite the weight-saving processing, which reduced the current collection ability (Supplementary Fig. 7). However, at higher operating rates, the capacity decreased in the laser-drilling electrode. This result may be due to the insufficient current collection ability based on the large hole size.

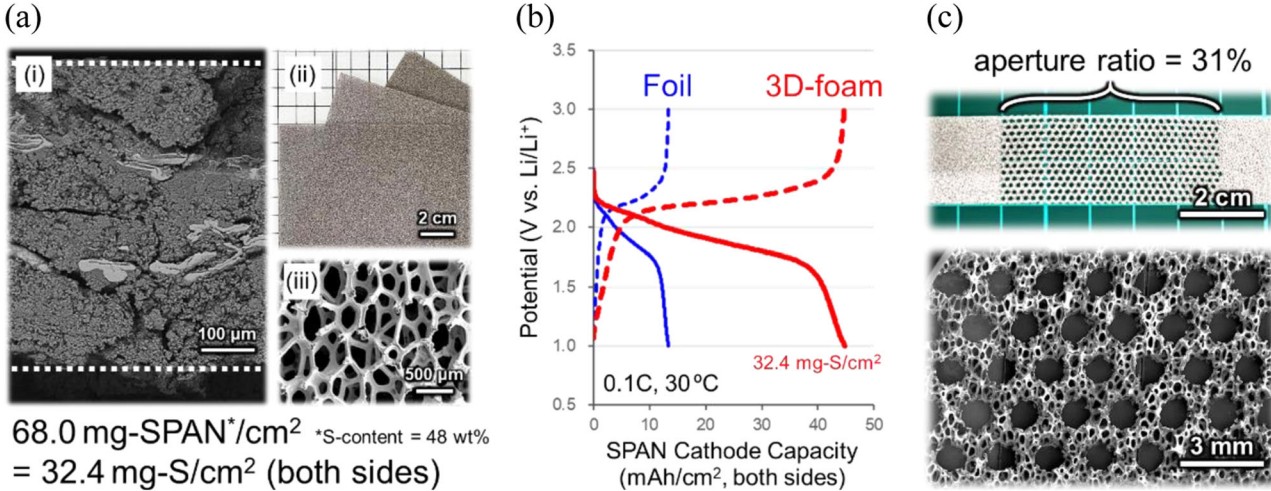

**Fig. 2 | Sulfurized polyacrylonitrile (SPAN) cathode and 3D-Al foam sheets as the current collector. a** (i) Cross-sectional scanning electron microscope (SEM) image of the thick SPAN layer in the 3D-Al foam with a density of 1.2 g cm$^{-3}$ and a porosity of 36% and (ii,iii) photographs of the 3D-Al foam (Al-CELMET). **b** Reversible chg./ dischg. properties at 0.1C-rate and 30 °C of the SPAN cathodes with carbon-coated Al foil (blue lines) or 3D-Al foam sheet (red lines) in pouch cells (SPAN cathodes | carbonate electrolyte solution | Li–metal anode). **c** Photographs of the porous 3D-Al foam sheet weight-saved by the laser-drilling technique.

**Table 1 | Correlation between SPAN cathodes with different compositions and thickness and reversible chg./dischg. capacities**

| Composition of SPAN Cathode | SPAN Layer | Capacity (mAh g$_{(SPAN)}^{-1}$) [0.1C-rate, 30 °C] |
|---|---|---|
| SPAN:AB:SBR:CMC-Na (92.0/5.0/1.5/1.5, wt%) | thin | 686 |
| SPAN:AB:SBR:CMC-Na (92.0/5.0/1.5/1.5, wt%) | thick | 685 |
| SPAN:AB:SBR:CMC-Na (94.0/3.0/1.5/1.5, wt%) | thin | 685 |
| SPAN:AB:SBR:CMC-Na (96.0/1.0/1.5/1.5, wt%) | thin | 651 |
| SPAN:MWCNT:SBR:CMC-Na (97.4/1.0/0.7/0.9, wt%) | thin | 674 |
| SPAN:SWCNT:SBR:CMC-Na (97.4/1.0/0.7/0.9, wt%) | thin | 685 |
| SPAN:SWCNT:SBR:CMC-Na (97.7/0.7/0.7/0.9, wt%) | thin | 686 |
| SPAN:SWCNT:SBR:CMC-Na (98.0/0.4/0.7/0.9, wt%) | thin | 685 |
| SPAN:SWCNT:SBR:CMC-Na (98.0/0.4/0.7/0.9, wt%) | thick | 673 |
| SPAN:SWCNT:SBR:CMC-Na:CNF (98.0/0.4/0.7/0.7/0.2, wt%) | thick | 685 |

**Fig. 3 | Sulfurized polyacrylonitrile (SPAN) cathodes with fiber or porous fiber. a** Scanning electron microscope (SEM) image of the SPAN cathode consisting of 90 wt% particle and 10 wt% fiber. **b** Nyquist plots at 30 °C of the SPAN cathodes with/without the SPAN fiber under a non-faradaic condition in symmetric coin cells (SPAN cathode | carbonate electrolyte solution with FEC | SPAN cathode) and theoretical simulation lines of non-faradaic/faradaic processes at porous electrodes. **c** SEM image of the porous SPAN fiber. **d** Chg./dischg. cycle performances at 0.3C-rate and 30 °C of thick SPAN cathodes in the 3D-Al foam with/without the SPAN fibers in pouch cells (SPAN cathodes | carbonate electrolyte solution | Li–metal anode).

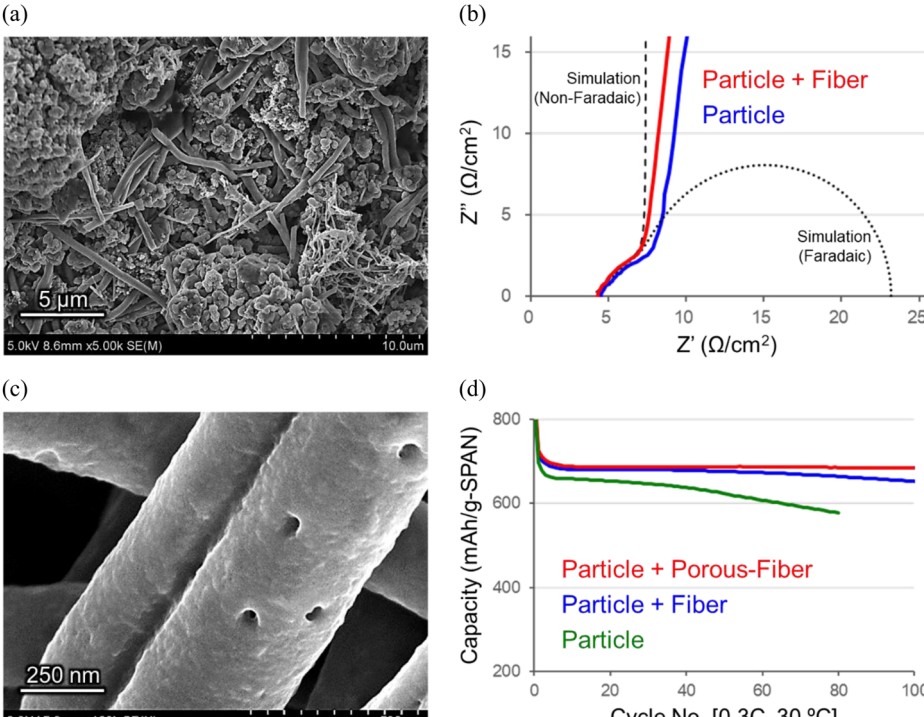

## Technology III

The active material weight ratio in the cathode layer is also important for energy density improvement[2]. Single-walled carbon nanotube (SWCNT) conducting agents and cellulose nanofiber (CNF) binders, which can be added in small quantities because of their superior electrical conductivity with high surface area and mechanical strength with the thixotropic property in water, are highly useful materials for this purpose[29,30]. A cathode fabricated using these materials with a 98.0 wt% SPAN ratio was designed, and it showed a favorable specific capacity (Table 1).

## Technology IV

SWCNTs are difficult to disperse owing to their strong cohesion, and as a result, ion diffusion in electrodes may deteriorate, instead of excellent electronic conductivity. A soft dispersing method, which disperses SWCNTs of good quality but does not cause damage, was selected, and Nihon Spindle Manufacturing's JET PASTER (JP) technique[31] was applied to SWCNT dispersion in H$_2$O. The SPAN cathode with the JP-treated SWCNT was found to have a low ohmic resistance ($R_{ohm}$) in the electrode (Supplementary Fig. 8).

## Technology V

Compared with conventional lithium transition metal oxides and sulfur–carbon composites as the cathode active materials, polymer-based materials provide a higher degree of freedom in terms of shape design[32]. Mixing active materials of particle and fiber shapes would improve electronic and ionic conductivities in the electrode. At the optimum blend ratio (90/10) of SPAN particles and fibers, the electronic conductivity remained approximately the same, but the ion diffusion resistance ($R_{ion}$) was reduced specifically in the SPAN cathode (Figs. 3a, b, and Supplementary Fig. 9). This result was analyzed by 3D microstructure simulations for models of SPAN cathodes and found to be due to the low-tortuosity pores for the lithium-ion transport path in the cathode (Supplementary Fig. 10).

## Technology VI

In the high-sulfur-loading thick cathodes demanded superior energy densities, electrolyte solutions may be depleted during chg./dischg. cycle operations. To improve the performance of electrolyte retention in the thick electrodes, a porous SPAN fiber was effective (Figs. 3c, 3d, and Supplementary Fig. 11). The application of the porous SPAN fiber with excellent

**Fig. 4 | Chg./dischg. characteristics of the sulfur-ized polyacrylonitrile (SPAN) cathode with expanded operations in coin cells (SPAN cathode | carbonate electrolyte solution with FEC | Li–metal anode). a** Reversible chg./dischg. properties at 0.1C-rate and 30 °C (red lines: 3.5–0.3 V, blue lines: 3.0–1.0 V vs. Li/Li$^+$) and a reaction potential between Al and Li. **b** Chg./dischg. cycle performances at 0.1 and 0.05C-rates and 30 °C after a formation process of ten cycles at 0.1C-rate and 30 °C.

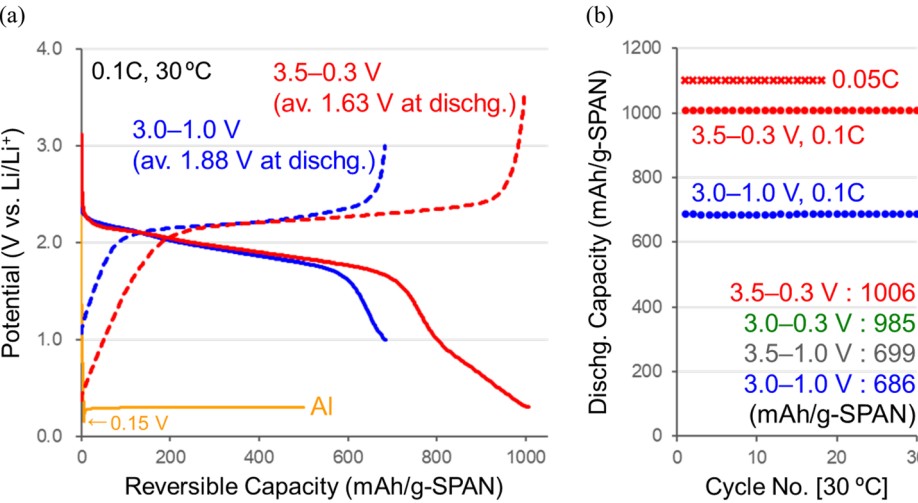

electrolyte solution absorbency was also effective in the cell assembly process (Supplementary Fig. 12).

## Technology VII

The chg./dischg. operation potential range of SPAN cathodes is limited within 3.0–1.0 V (*vs.* Li/Li$^+$) to utilize the sulfur redox reaction[13–20]. On the other hand, the cyclized-PAN backbones were electrochemically active[33] and had reversible capacities at the lower potentials (<1.0 V). An expanded chg./dischg. operation was performed in a potential range of 3.5–0.3 V to prevent undesirable side reactions of the Al current collector and carbonate/ether electrolyte solutions[34,35]. As a result, higher specific capacities of 1006 mAh g$_{(SPAN)}^{-1}$ (2113 mAh g$_{(S)}^{-1}$) with 100% coulombic efficiency at 0.1C-rate (100 mA g$_{(SPAN)}^{-1}$) and 1102 mAh g$_{(SPAN)}^{-1}$ (2315 mAh g$_{(S)}^{-1}$) at 0.05C-rate (50 mA g$_{(SPAN)}^{-1}$) were observed after ten cycles at 30 °C (Fig. 4a). Figure 4b shows chg./dischg. cycle performances at 0.1 and 0.05C-rates and 30 °C after a formation process of ten cycles at 0.1C-rate and 30 °C. Stable chg./dischg. cycle operations were confirmed even when the potential range was expanded. Further cycle stabilities are being tested.

## Technology VIII

In general, 30–60% of the Li–S pouch cell weight is attributable to the electrolyte solutions; therefore, reducing the weight of the solutions is essential for improving the gravimetric energy density[2,6,9,36–38]. A new ether-based electrolyte solution (Light-Ele) with the combined properties of lightweight (0.98 g cm$^{-3}$), high ionic conductivity, and low viscosity was developed and confirmed to work in the Li–SPAN cell (Supplementary Table 2 and Supplementary Fig. 13). However, the activity of SPAN with the Light-Ele was lower than those with the conventional carbonate-based electrolyte solutions, and we inferred that the quality of a cathode–electrolyte–interphase (CEI) film was poor in the case of the Light-Ele. Our idea to solve this problem is to apply a two-step chg./dischg. method using two different electrolytes. In the first step, a carbonate-based electrolyte solution with FEC and LiBOB additives is used to form a suitable CEI film[39,40], and in the second step, ether-based Light-Ele is used to reduce cell weight after removing the electrolyte used in the first step. The two-step method was successfully applied, resulting in sufficient SPAN performance in Light-Ele (Table 2). This new technique is unique to SPAN, which can operate with a variety of electrolyte solutions. Furthermore, SPAN can reduce the amount of electrolyte solutions in the cells because the elution of sulfur components into electrolyte solutions hardly occurs, and this two-step method enabled further reductions.

## Technology IX

A thinner separator is also essential for increasing the cell energy density, but Li–S cells are prone to short circuits when a thin separator is used. The

SETELA PE-type separator film, which is 5 μm thick and has 35% porosity favorable for lightweight cells, showed a good affinity for Li–SPAN cells (Supplementary Table 3). In addition, a thinner pouch of an aluminum laminated film with a thickness under 80 μm (thin-type DNP Battery Pouch) was also applied.

## Technology X

To maximize the energy density performance of Li–metal anode cells, anode-free configuration designs, that is, lithium deposits on the bare current collector without any host materials, are useful, and there have been many reports on cells with Li-containing cathode active materials[41–46]. Since SPAN does not contain the Li element, an electrochemical prelithiation strategy using the half-cell method was applied to the SPAN cathode (Supplementary Fig. 14). The carbonate-based electrolyte solution with FEC and LiBOB additives was used for the prelithiation and the ether-based

**Table 2 | Correlation between electrolyte solutions and cell properties**

| Electrolyte Solution (chg./dischg. cycle number at 0.1C-rate and 30 °C) | Normalized Capacity after 20 cycles | Normalized Resistance after 20 cycles |
|---|---|---|
| Ele-1 (20) | 1.00 | 1.02 |
| Ele-2 (20) | 1.00 | 1.01 |
| Ele-3 (20) | 1.00 | 1.00 |
| Ele-4 (20) | 0.98 | 1.05 |
| Ele-5 (20) | 0.89 | 1.10 |
| Ele-6 (20) *Ele-6 = Light-Ele | 0.95 | 1.07 |
| Ele-1 (5) → Ele-6 (20) [two-step method] | 0.97 | 1.05 |
| Ele-2 (5) → Ele-6 (20) [two-step method] | 0.99 | 1.02 |
| Ele-3 (5) → Ele-6 (20) [two-step method] | 1.00 | 1.00 |
| Ele-3 (5) → Ele-4 (20) [two-step method] | 1.00 | 1.02 |
| Ele-3 (5) → Ele-5 (20) [two-step method] | 0.98 | 1.03 |

Ele-1: 1.0 M LiPF$_6$ in EC/DEC (50/50, vol%).
Ele-2: 1.0 M LiPF$_6$ in FEC/DEC (50/50, vol%).
Ele-3: 1.0 M LiPF$_6$ in FEC/DEC (50/50, vol%) + 2 wt% LiBOB.
Ele-4: 1.0 M LiTFSI in DOL/DME (50/50, vol%) + 2 wt% LiNO$_3$.
Ele-5: 0.4 M LiTFSI + 0.4 M LiNO$_3$ + 0.1 M LiHFDF in DME/DOL/TFMTMS (48/17/35, vol%).
Ele-6: 0.2 M LiTFSI + 0.2 M LiFSI + 0.1 M LiNO$_3$ + 0.1 M LiHFDF in DME/DOL/TFMTMS (75/5/20, vol%).

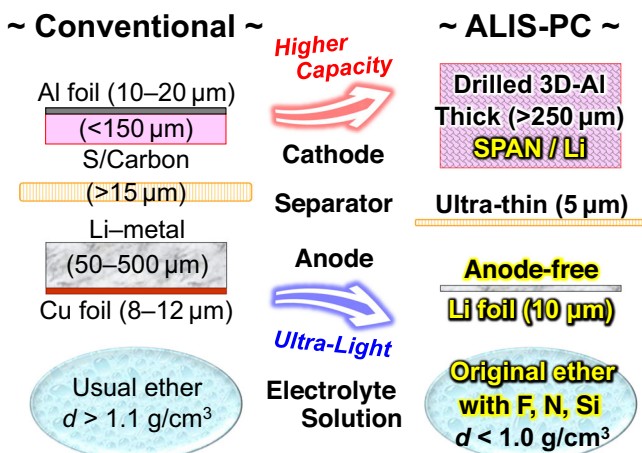

**Fig. 5 | ADEKA's Lithium–Sulfur/Pouch Cell (ALIS-PC) design with the sulfurized polyacrylonitrile (SPAN) cathode for an ultra-lightweight Li–S rechargeable battery cell.** The left graphic shows a conventional lithium–sulfur (Li–S) constitution design. The right graphic shows the state-of-the-art ultra-lightweight Li–S constitution design, that is ALIS-PC, developed in this work. The ALIS-PC design is well suited to our SPAN (ADEKA AMERANSA SAM-8).

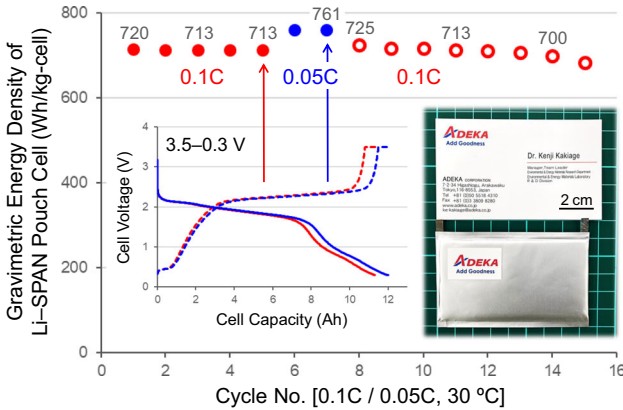

**Fig. 6 | Chg./dischg. cycle performance of the ultra-lightweight lithium–sulfurized polyacrylonitrile (Li–SPAN) pouch cell with the ADEKA's Lithium–Sulfur/Pouch Cell (ALIS-PC) design.** 0.1C (red lines) and 0.05C (blue lines)-rates properties with 3.5–0.3 V operation at 30 ºC, chg./dischg. characteristic curves at 713 and 761 Wh kg$_{(cell)}$$^{-1}$, and a photograph of the Li–SPAN pouch cell.

Light-Ele for the completed cell with the lithiated SPAN cathode and the anode-free configuration design. In other words, the two-step chg./dischg. method with two different electrolyte solutions described above was well applied to the anode-free-type Li–SPAN cell. The low coulombic efficiency and Li-dendrite deposition straiten a stable cell operation in the anode-free designs. We solved the problems by applying an ultra-thin Li foil of ca. 10 μm thickness as the negative current collector instead of the conventional Cu foil and also by using the electrolyte additive LiHFDF[47] in the Light-Ele. The 100% coulombic efficiency allowed for a reduction in the amount of electrolyte solution, and SPAN is not limited by the E/S ratio, unlike other sulfur-based active materials[11,12,48,49]. In addition, lithium has a significantly lower density (0.53 g cm$^{-3}$) than copper (8.96 g cm$^{-3}$); therefore, this design is an ideal material choice for ultra-lightweight cells. Figure 5 shows an original Li–S pouch cell design that consolidates the state-of-the-art technologies described above, which is named ALIS-PC (ADEKA's Lithium–Sulfur/Pouch Cell). SPAN is probably the most suitable cathode active material for the ALIS-PC design.

## Ultra-lightweight cell

An ultra-lightweight Li–SPAN pouch cell with the ALIS-PC design realized by applying the above ten technologies was assembled with an 11 Ah-class capacity (Supplementary Table 4). The electrode density of the SPAN cathode in this work was 1.2 g cm$^{-3}$ with 36% porosity. It is the eminently low porosity among sulfur-based cathodes of Li–S batteries, but it is the high porosity compared to NCM cathodes of usual LIBs. We have therefore developed the lightweight electrolyte solution with the low density of less than 1.0 g cm$^{-3}$. As a result, the proportion of the electrolyte solution in the overall weight of the ultra-lightweight designed pouch cell could be controlled to 28%. The two-step method of Technology VIII and the porous SPAN fiber of Technology VI were also important for the development of the Li–SPAN pouch cell that initial-operate stably with smaller amounts of the lightweight electrolyte solution. There are probably no examples of electrolyte weight percentages below 30% in well-operating Li–S pouch cells. Figure 6 and Supplementary Fig. 15 show chg./dischg. cycle performance and characteristic curves in five cycles at 0.1C-rate (100 mA g$_{(SPAN)}$$^{-1}$, 68.0 mg$_{(SPAN)}$ cm$^{-2}$ on both sides) and 30 °C. An outstanding discharge performance at 11.31 Ah and 1.64 V was observed in the 26.03 g$_{(cell)}$ (without fixtures), and the gravimetric energy density based on the total mass of all cell components was calculated to be 713 Wh kg$_{(cell)}$$^{-1}$ (volumetric; 832 Wh L$_{(cell)}$$^{-1}$). At a lower rate operation of 0.05 C (50 mA g$_{(SPAN)}$$^{-1}$), the energy densities were 761 Wh kg$_{(cell)}$$^{-1}$ and

889 Wh L$_{(cell)}$$^{-1}$ (12.01 Ah and 1.65 V) and 800 Wh kg$^{-1}$ excluding the weights of the pouch film and metal tabs. The characteristic curves at 0.05C-rate is also shown in Fig. 6. When the thickness of the Li–metal anode increased, the thickness of the SPAN cathode reduced to the same degree. As a result, the total volume change of the ultra-lightweight Li–SPAN pouch cell during the initial chg./dischg. cycles was very small and the stable initial operation was successfully achieved. The number of chg./dischg. cycles and the C-rate characteristics can be improved by reducing the cell energy density (Supplementary Fig. 16), and it is important to design appropriately for each application. The ultra-lightweight Li–SPAN pouch cell in this work could be used below 0.2C-rate. The C-rate performance is relatively good, as lightweight Li–S cells to date have often been limited to 0.05C-rate or less.

In conclusion, by fabricating the world's lightest rechargeable battery cell with >750 Wh kg$_{(cell)}$$^{-1}$ through the fusion of various chemistries and chemical engineering (Supplementary Fig. 17), we clearly demonstrated the potential of Li–SPAN designs to approach innovative post-LIB realizations (Fig. 1 and Supplementary Table 1). Recently, a wide variety of rechargeable batteries have been in demand, and this study can provide some of the most important results to meet the demand. We envision a future in which the commercialization and practical application of Li–SPAN batteries will exploit new markets and accelerate the development of a sustainable and prosperous society.

## Methods
### Preparation of SPAN cathodes

The aqueous slurries for the cathode layer coating were prepared by mixing sulfurized polyacrylonitrile particle [SPAN: ADEKA AMERANSA SAM-8/ 48 wt% sulfur content/D$_{50}$ = 3 μm/particle density = 1.9 g cm$^{-3}$/tap density = ca. 0.6 g cm$^{-3}$; ADEKA CORPORATION (Supplementary Fig. 18)] as the active material with acetylene black (AB; DENKA BLACK Li-100, Denka Company Limited), multi-walled carbon nanotube (MWCNT; FT9000, Jiangsu Cnano Technology Co., Ltd.), or single-walled carbon nanotube (SWCNT; Lamfil WPB-043/H$_2$O dispersion, Kusumoto Chemicals, Ltd.) as conductive agents and styrene–butadiene rubber (SBR; BM-451B/water-based, Zeon CORPORATION) and sodium carboxymethyl cellulose (CMC-Na; CELLOGEN BSH-6, DKS Co., Ltd. or DAICEL CMC 2200, Daicel Miraizu Ltd.) and cellulose nanofiber (CNF; under development, ADEKA CORPORATION) as binders at SPAN:AB:SBR:CMC-Na weight ratios of 92.0/5.0/1.5/1.5, 94.0/3.0/1.5/1.5, and 96.0/1.0/1.5/1.5; or SPAN:MWCNT:SBR:CMC-Na weight ratio of 97.4/1.0/0.7/0.9; or SPAN:SWCNT:SBR:CMC-Na weight ratios of 97.4/1.0/0.7/0.9, 97.7/0.7/ 0.7/0.9, and 98.0/0.4/0.7/0.9; or SPAN:SWCNT:SBR:CMC-Na:CNF weight ratio of 98.0/0.4/0.7/0.7/0.2. The dispersion of SWCNT in H$_2$O using the

JET PASTER JPSS-X was carried out by Nihon Spindle Manufacturing Co., Ltd. SPAN fiber and porous fiber with ca. 500 nm diameter were synthesized as follows. 10 wt% polyacrylonitrile (PAN) and PAN/poly(methyl methacrylate) (90/10 by weight, Merck KGaA) solutions were used as spinning solutions. A predetermined amount of polymer powder was dissolved in N,N-dimethylformamide (99.5%, Nacalai Tesque Inc.) by stirring at room temperature (RT) for 2 h and then at 60 °C for 6 h, which was followed by slow cooling to RT. The prepared solution was electrospun using a NEU Nanofiber Electrospinning Unit (KATO TECH CO., LTD.). The voltage and flow rate were fixed at 15 kV and 1.0 mL h$^{-1}$, respectively. The inner diameter of the nozzle (stainless-steel needle) and the distance between the needle tip and the rotating collector were 0.94 mm and 100 mm, respectively. The collector, which was covered with aluminum foil, was rotated at a speed of 1 m min$^{-1}$. The temperature and ambient humidity throughout the process were 25 °C and 25%, respectively. The obtained electrospun polymer fibers were peeled off from the aluminum foil. The polymer fibers and excess elemental sulfur ($S_8$, Hosoi Chemical Industry Co., Ltd.) were mixed by using a mortar and pestle, and the mixtures were maintained at 300–500 °C in a nitrogen atmosphere for thermal conversion reactions. Subsequent heat treatments removed the elemental sulfur, which was confirmed by powder XRD analyses (Supplementary Fig. 19) using Cu-Kα radiation at 40 kV and 40 mA (Ultima IV, Rigaku Corporation). An elemental analysis confirmed a sulfur content of 48 wt% in the SPAN fibers, and the fibers became porous when using PAN/poly(methyl methacrylate) as the raw material. Portions of the SPAN particle (SAM-8) in the cathodes were replaced by the SPAN fibers. Subsequently, the obtained aqueous SPAN slurries were coated onto carbon-coated aluminum (Al) foil (SDX, Showa Denko K.K.) or 3D-Al foam sheet (Al-CELMET, 1.0 mm thick/96% porosity/10 mg cm$^{-2}$, Sumitomo Electric Industries, Ltd.) as cathode current collectors and dried in ovens at 80 °C for 8 h or 70 °C for 15 h, respectively. The laser drilling of the 3D-Al foam for weight saving was carried out by WIRED Co., Ltd. The SPAN electrode with the weight saved 3D-Al foam possessing a thickness of 578 µm, a SPAN loading of 68.0 mg cm$^{-2}$, a density of 1.2 g cm$^{-3}$, and 36% porosity was obtained using roll press equipment (Oono-roll Corporation). Lastly, the cathodes were shaped into 12-φ disc-size for coin-type cells or 8.0 × 4.2 cm$^2$ rectangle-size for pouch-type cells, and dried in a vacuum oven at 130 °C for 15 h before cell assembling.

### Preparation of electrolyte solutions

1.0 M LiPF$_6$ in EC/DEC (50/50, vol%), 1.0 M LiPF$_6$ in FEC/DEC (50/50, vol%), 1.0 M LiPF$_6$ in FEC/DEC (50/50, vol%) + 2 wt% LiBOB, 1.0 M LiTFSI in DOL/DME (50/50, vol%) + 2 wt% LiNO$_3$, 0.4 M LiTFSI + 0.4 M LiNO$_3$ + 0.1 M LiHFDF in DME/DOL/TFMTMS (48/17/35, vol%)[38], and Light-Ele of 0.2 M LiTFSI + 0.2 M LiFSI + 0.1 M LiNO$_3$ + 0.1 M LiHFDF in DME/DOL/TFMTMS (75/5/20, vol%) were prepared in an Ar-filled glove box. The water contents in the solutions were controlled to be under 30 ppm. Electrolyte solutions of 1.0 M LiPF$_6$ in EC/DEC (50/50, vol%) and 1.0 M LiPF$_6$ in FEC/DEC (50/50, vol%) were purchased from KISHIDA CHEMICAL CO., LTD. Lithium bis(fluorosulfonyl)imide LiN(SO$_2$F)$_2$ (LiFSI; IONEL LF-101) was provided by NIPPON SHOKUBAI CO., LTD. Lithium nitrate (LiNO$_3$) was purchased from FUJIFILM Wako Pure Chemical Corporation, and other materials such as lithium bis(oxalate)borate (LiBOB), 1,3-dioxolane (DOL), 1,2-dimethoxyethane (DME), lithium 1,1,2,2,3,3-hexafluoropropane-1,3-disulfonimide (LiHFDF), (trifluoromethyl)trimethylsilane (TFMTMS), and lithium bis(trifluoromethanesulfonyl)imide (LiTFSI) were purchased from Tokyo Chemical Industry Co., Ltd.

### Cell assemblies

Coin-type and pouch-type cells were assembled in dry spaces with dew points between −50 and −75 °C using a Dry Room System (Seibu Giken Co., Ltd.) and an NS DRY BOOTH (Nihon Spindle Manufacturing Co., Ltd.). The compositions of 2032 coin-type cells were as follows: SPAN cathodes (12φ) | Celgard 2325 and ADVANTEC GA-100 separators (16φ)| Li–metal anode without Cu foil (14φ/500 µm thick, Honjo Metal Co., Ltd.) with filled electrolyte solutions. The coin-cell pieces were purchased from Hohsen Corp. In the case of ultra-lightweight pouch-type cells, 5 sheets of the SPAN cathodes with the 3D-Al foam (8.0 × 4.2 cm$^2$), 10 sheets of a SETELA-05C separator (8.4 × 4.6 cm$^2$/PE/5 µm thick/35% porosity, Toray Industries, Inc.), and 6 sheets of a Li–metal anode current collector without Cu foil (8.7 × 4.4 cm$^2$/ca. 10 µm thick, Honjo Chemical Corporation) were alternately stacked, and then an Al tab for the cathode, a Ni-tab for the anode, and Al-laminated film (DNP Battery Pouch/D-EL30H(3)20/77 µm thick/0.013 g cm$^{-2}$, Dai Nippon Printing Co., Ltd.) were attached to the stacked electrodes. After electrolyte injection, the pouch cells were heat-sealed and ready for battery operations. The E/S ratio in the case of the Light-Ele was 1.4 µL mg$^{-1}$. A 500-µm-thick Li–metal anode with a Cu foil (10 µm thick), excess carbonate electrolyte solutions in large amounts, and Celgard 2325 were used for the electrochemical prelithiation of the SPAN cathodes for five cycles at 0.1C-rate and 30 °C (Supplementary Fig. 14), and the prelithiated cathodes were washed by using dehydrated dimethyl carbonate. All pouch-type cells were strongly constrained with fixtures in chg./dischg. operations.

### Cell operations and analyses

Charge-and-discharge (chg./dischg.) measurements were performed by using the chg./dischg. test systems TOSCAT (TOYO SYSTEM CO., LTD.) and BLS (KEISOKUKI CENTER CO., LTD.) at 30 °C. The cut-off voltages of 3.0/1.0, 3.0/0.3, 3.5/1.0, and 3.5/0.3 V were set for chg./dischg. steps. CCCV-chg. (CV time = 30 min.)/CC-dischg. and CC-chg./CC-dischg. modes were used in 11 Ah-class pouch cells and other cells, respectively. Electrochemical impedance spectroscopy (EIS) measurements using Solartron 1286 Potentiostat/Electrochemical Interface and Solartron 1260 Impedance/Gain-Phase Analyzer (AMETEK, Inc./Solartron Analytical) were carried out in the frequency range from 100 kHz to 0.1 Hz with a sinusoidal AC voltage amplitude of 10 mV. The EIS data were analyzed by using the ZView software program. Regulus8220 (Hitachi High-Tech Corporation), Ultim Max 100 (Oxford Instruments KK), and Ultra-microtome Leica EM UC7 (Leica Microsystems GmbH) were used for field emission scanning electron microscope (FE-SEM) observation and energy-dispersive X-ray spectroscopy (EDX) elemental mapping. Tortuosity simulations of the SPAN electrode models with/without the SPAN fiber were performed using the GeoDict software (Math2Market GmbH)[50], and calculations were carried out by NISSAN ARC, LTD.

### Data availability

All data generated or analyzed during this study are included in this published article (and its supplementary information files).

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

## Acknowledgements

We received generous support from the "National Institute of Advanced Industrial Science and Technology (AIST), SENOH Group at AIST Kansai, Japan" and "ATTACCATO LLC., Japan".

## Author contributions

K.K. conceived the research idea, managed the project, designed and performed the experiments, wrote the original manuscript, and revised the manuscript. T.Y. and H.U. contributed to the discussion of the data. M.K. provided the ideas and methods for the materials, performed the experiments, and contributed to the discussion of the data and revision of the manuscript.

## Competing interests

The authors declare the following competing interests: This study was funded by ADEKA CORPORATION. K.K. and T.Y. are employees of ADEKA CORPORATION.
