## [Peer Review File · Communications Engineering]

Ultra-lightweight rechargeable battery with enhanced gravimetric energy densities >750 Wh kg⁻¹ in lithium–sulfur pouch cellReviewers' comments:

Reviewer #1 (Remarks to the Author):

In this manuscript, a Li-SPAN pouch cell was fabricated with an ultra-high energy density of 761 Wh/kg. The authors employed multiple approaches to reduce the contents of inactive materials in the pouch cell and improved the battery performance by building a superior electron network. Some of these strategies lead to quite promising results for the future development of high-energy-density Li-S batteries. This work could be published on Communications Engineering pending satisfactory revisions.

1. What is the thickness of 3D Al foam? And what about its mechanical property after introducing the SPAN material? Or simply, can it be bent or folded?
2. The author employed electrochemical pre-lithiation technology on the cathode. Does this technology fit large-scale production?
3. During the charging process, Li ions will be liberated and form metal deposits on the anode, leading to an increased thickness of about 150 μm on each side of the anode (corresponding to 30 mAh/cm²). Can the pouch cell accommodate such significant volume expansion? Additionally, what is the volumetric energy density of pouch cell at the charged state?
4. What is the expected material cost of this pouch cell? Is it comparative with current lithium-ion batteries?
5. Among the pouch cells with varying energy densities depicted in Supplementary Figure 16, those with lower energy density show better cycling performance, while their counterparts with higher energy density exhibit declined cycling performance. In the context of commercial applications, which parameters should be considered to strike the optimal balance?

Reviewer #2 (Remarks to the Author):

The improved performance and commercialization of Li-S batteries will enrich our lives with EV revolution and urban air mobility. This work designed the world's lightest rechargeable battery cell by combining the sulfurized polyacrylonitrile cathode and state-of-the-art technologies involving chemical engineering. The highest gravimetric energy densities of these Li-S pouch cells were 713 and 761 Wh/kg-cell after some charge-and-discharge cycles, which significantly exceeded those of commercial lithium-ion and developed next-generation rechargeable batteries. The pouch cell performance achieved in this work is commendable. This work also provided a series of techniques to achieve the final performance by optimizing multiple components such as the cathode, anode, separator, electrolyte, conductive agent, and charging and discharging parameters of the pouch cell. However, the authors are suggested to address the following questions before the possible acceptance for publication.

1. In this manuscript, the author claims that applying SPAN (ADEKA AMERANSA SAM-8) and ten state-of-the-art technologies can realize the 713 and 761 Wh/kg gravimetric energy density of lithium-sulfur pouch cells. What are the volumetric energy densities of these pouch cells, and at what level? Given that these cathodes have high porosity, could that lead to low volumetric energy densities? What is the tap density?

2. In Supplementary Fig. 7, the cells with laser drilling exhibit lower rate performance (0.3 C and 0.5 C) compared to cells without laser drilling. Specific analysis or reasons should be provided.
3. The author actually achieves the final cell performance by optimizing multiple components such as the cathode, anode, separator, electrolyte, conductive agent, and charging and discharging parameters of the pouch cell. But the Technology I II III...-form discussion reads more like stacked technical reports. The manuscript reads like a patent or technical report. Although the Communications Engineering journal did not require a specific format during the initial review, I still suggest that the author can make revisions to the style of writing and formatting. The pouch cell performance achieved in this work is commendable. Therefore, I suggest that the author should spend time to make the results and discussion sections more coherent and logical. Good writing will bring the work to the attention of more readers.
4. In the Technology VIII, 30–60% of the Li–S pouch cell weight is attributable to the electrolyte solutions. Therefore, reducing the weight of the solutions is essential for improving the gravimetric energy density. Some studies believe that higher porosity in Li–S cathodes means more electrolyte is needed to fill it. Some cathodes have high porosity in this manuscript, how do the authors address this?
5. Have the authors considered using an electrolyte without adding lithium nitrate? This may help the authors to be more confident about the ability of the SPAN-based material to suppress the shuttle effect.
6. This manuscript contains rich TEM and SEM data, but the corresponding analysis and discussion are not sufficient. It is recommended to refer to the following two works: *Scripta Materialia*, 2020, 187:107-112. *Corrosion Science*, 2019, 152: 54-59.
7. In response to the problems of Li–S cells, many reports have introduced a variety of excellent adsorbent materials and catalytic materials to adsorb and catalytically convert polysulfides. Most of them have achieved notable progress in coin cells. But why is it still difficult to improve the energy density of Li–S pouch cells? What are the key issues or problems in this? Is it about materials or technologies, or both? The introduction should include more to highlight the importance and uniqueness of the work.
8. It is recommended to perform fitting analysis on the EIS results in the manuscript.

Reviewer #3 (Remarks to the Author):

This paper reported a new designed ultra-light weight Li–S pouch cell, using Sulfurized polyacrylonitrile as cathode active material combining with ten state-of-art technologies, achieved the highest gravimetric energy density of 761Wh/kg, which is very promising for the next-generation batteries. The manuscript can be considered for publication. Below are my comments for the authors,

- 1) The research combines ten state-of-art technologies to achieve the goal, would the authors know which technologies are the most important factors? Have the authors considered or tried to fabricate a baseline cell and fabricate cells only with one technology to compare? It would be interesting to see the differences caused by single factor.
- 2) This paper only mentioned cycling tests using very low C rate under 30 degC, how about its performance in fast charging or low temperature?
- 3) What would be the difficulties for the new cell towards mass production? What are the future plans from the authors?

Specifically,

- 4) Wonder why the author chose to do the cycling tests under 30 degC, not 25 degC, 40 degC or other temperature points?
- 5) In Technology VII part, is 'cyclized-PAN' a typo? Should it be 'cyclized-SPAN'?
- 6) In Technology VIII part, why the quality of the CEI film was poor in the case of the Light-Ele?
- 7) In Ultra-lightweight cell part, from Supplementary Fig. 16 it showed different charge/discharge C rates were used, and the voltage window in cycling tests were also different, is it appropriate to get the conclusion 'The number of chg./dischg. cycles can be improved by reducing the cell energy density'?

Reviewer #1 (Remarks to the Author):

In this manuscript, a Li-SPAN pouch cell was fabricated with an ultra-high energy density of 761 Wh/kg. The authors employed multiple approaches to reduce the contents of inactive materials in the pouch cell and improved the battery performance by building a superior electron network. Some of these strategies lead to quite promising results for the future development of high-energy-density Li-S batteries. This work could be published on Communications Engineering pending satisfactory revisions.

Answer: Thank you for your positive review for our study. Your comments were very helpful in the development of our study.

1. What is the thickness of 3D Al foam? And what about its mechanical property after introducing the SPAN material? Or simply, can it be bent or folded?

Answer: The thickness of the 3D-Al foam sheet is 1.0 mm (See Methods section). The SPAN cathode with the 3D-Al foam sheet has moderately flexible mechanical properties and can be bent but not folded.

2. The author employed electrochemical pre-lithiation technology on the cathode. Does this technology fit large-scale production?

Answer: The electrochemical prelithiation technology of this paper was carried out the using lab-scale half-cell method. Industrial prelithiations in large-scale are employed in lithium-ion capacitors. Roll-to-roll prelithiation technology of LIB's electrodes has been developed, for example, by Musashi Energy Solutions Co., Ltd.

3. During the charging process, Li ions will be liberated and form metal deposits on the anode, leading to an increased thickness of about 150 μm on each side of the anode (corresponding to 30 mAh/cm²). Can the pouch cell accommodate such significant volume expansion? Additionally, what is the volumetric energy density of pouch cell at the charged state?

Answer: Thank you for your comment. This is an important point. When the thickness of the Li-metal anode increases, the thickness of the SPAN cathode reduces to the same degree. As a result, the total volume expansion of the Li-SPAN pouch cell during initial chg./dischg. cycles is very small. The explanation was added in the revised manuscript.

4. What is the expected material cost of this pouch cell? Is it comparative with current lithium-ion batteries?

Answer: This pouch cell fabrication is lab-scale and therefore not cost-compatible with

mass-produced current lithium-ion batteries. Future scale-up will provide cost projections.

5. Among the pouch cells with varying energy densities depicted in Supplementary Figure 16, those with lower energy density show better cycling performance, while their counterparts with higher energy density exhibit declined cycling performance. In the context of commercial applications, which parameters should be considered to strike the optimal balance?

Answer: The required optimal balance between gravimetric energy density and cycle lifespan varies depending on commercial applications. Battery cell parameters need to be designed according to the applications and this paper is useful for the designs.

Reviewer #2 (Remarks to the Author):

The improved performance and commercialization of Li–S batteries will enrich our lives with EV revolution and urban air mobility. This work designed the world's lightest rechargeable battery cell by combining the sulfurized polyacrylonitrile cathode and state-of-the-art technologies involving chemical engineering. The highest gravimetric energy densities of these Li–S pouch cells were 713 and 761 Wh/kg-cell after some charge-and-discharge cycles, which significantly exceeded those of commercial lithium-ion and developed next-generation rechargeable batteries. The pouch cell performance achieved in this work is commendable. This work also provided a series of techniques to achieve the final performance by optimizing multiple components such as the cathode, anode, separator, electrolyte, conductive agent, and charging and discharging parameters of the pouch cell. However, the authors are suggested to address the following questions before the possible acceptance for publication.

Answer: Thank you for your positive review for our study. Your comments were very helpful in the development of our study.

1. In this manuscript, the author claims that applying SPAN (ADEKA AMERANSA SAM-8) and ten state-of-the-art technologies can realize the 713 and 761 Wh/kg gravimetric energy density of lithium–sulfur pouch cells. What are the volumetric energy densities of these pouch cells, and at what level? Given that these cathodes have high porosity, could that lead to low volumetric energy densities? What is the tap density?

Answer: The energy densities of the ultra-lightweight Li–SPAN pouch cell were 713 Wh/kg-cell & 832 Wh/L-cell at 0.1C-rate and 761 Wh/kg-cell & 889 Wh/L-cell at 0.05C-rate, respectively. We have revised the main text. The Li–SPAN cell was also superior to the usual LIBs in terms of the volumetric energy density. A tap density of the SPAN powder is ca. 0.6 g/cm³, we have revised the main text.

2. In Supplementary Fig. 7, the cells with laser drilling exhibit lower rate performance (0.3 C and 0.5 C) compared to cells without laser drilling. Specific analysis or reasons should be provided.

Answer: Thank you for valuable discussion. The hole size of the laser-drilling treatment is large, $\phi = 1.0$ mm. This is considered to have reduced the C-rate characteristic due to the low current collection ability. The explanation was added in the revised manuscript. It is thought that the C-rate performance can be improved by smalling the laser-drilled hole size, and efforts are planned to achieve this.

3. The author actually achieves the final cell performance by optimizing multiple components such

as the cathode, anode, separator, electrolyte, conductive agent, and charging and discharging parameters of the pouch cell. But the Technology I II III...-form discussion reads more like stacked technical reports. The manuscript reads like a patent or technical report. Although the Communications Engineering journal did not require a specific format during the initial review, I still suggest that the author can make revisions to the style of writing and formatting. The pouch cell performance achieved in this work is commendable. Therefore, I suggest that the author should spend time to make the results and discussion sections more coherent and logical. Good writing will bring the work to the attention of more readers.

Answer: Thank you for your valuable comments on how to make the paper better. The aim of this paper is to share each original engineering and chemical technique and to make these techniques definitely useful for reader's research and developments. For this reason, we dared to choose the technical report-like form. After serious consideration of your advice, we would like to proceed with the paper in this format in order to achieve the above aim. The next updated results based on this paper would be reported in the proposed more coherent and logical form.

4. In the Technology VIII, 30–60% of the Li–S pouch cell weight is attributable to the electrolyte solutions. Therefore, reducing the weight of the solutions is essential for improving the gravimetric energy density. Some studies believe that higher porosity in Li-S cathodes means more electrolyte is needed to fill it. Some cathodes have high porosity in this manuscript, how do the authors address this?

Answer: Thank you for your comment. This is an important point. The electrode density of the SPAN cathode in this work was 1.2 g/cm^3 with 36% porosity. It is the eminently low porosity among sulfur-based cathodes of Li–S batteries, but it is the high porosity compared to NCM cathodes of usual LIBs. We have therefore developed the lightweight electrolyte solution with the low density of less than 1.0 g/cm^3 . As a result, the proportion of the electrolyte solution in the overall weight of the ultra-lightweight designed pouch cell could be controlled to 28%. The two-step method of Technology VIII and the porous SPAN fiber of Technology VI were also important for the development of the Li–SPAN pouch cells that operate stably with smaller amounts of the lightweight electrolyte solution. There are probably no examples of the electrolyte weight percentages below 30% in lightweight Li–S battery cells that operate stably for more than 10 cycles in chg./dischg. The explanation was added in the revised manuscript. The detailed design of the weights is shown in Supplementary Table 4, and we add a pie chart on weight distribution.

5. Have the authors considered using an electrolyte without adding lithium nitrate? This may help the authors to be more confident about the ability of the SPAN-based material to suppress the shuttle effect.

Answer: Electrolyte solutions without lithium nitrate were also tested in Li–SPAN cells; SPAN was less prone to the typical polysulfide redox shuttle phenomenon, so the effect of the lithium nitrate addition was much smaller than for other Li–S cells. A small amount of lithium nitrate was added to Light-Ele (Ele-6) in this work as it also has the effect of stabilizing the Li–metal anode.

6. This manuscript contains rich TEM and SEM data, but the corresponding analysis and discussion are not sufficient. It is recommended to refer to the following two works: *Scripta Materialia*, 2020, 187:107-112. *Corrosion Science*, 2019, 152: 54-59.

Answer: No TEM data exist for this manuscript. We plan to publish a paper on TEM analyses of SPAN electrodes in the future. The recommended papers are very useful as references when preparing that paper. Thank you very much for your comments and recommendation.

7. In response to the problems of Li-S cells, many reports have introduced a variety of excellent adsorbent materials and catalytic materials to adsorb and catalytically convert polysulfides. Most of them have achieved notable progress in coin cells. But why is it still difficult to improve the energy density of Li–S pouch cells? What are the key issues or problems in this? Is it about materials or technologies, or both? The introduction should include more to highlight the importance and uniqueness of the work.

Answer: Thank you for your valuable comments. The introduction has been revised according to the advice.

8. It is recommended to perform fitting analysis on the EIS results in the manuscript.

Answer: For non-faradaic process at porous electrodes, the overall impedance is expressed as a following equation [*J. Electrochem. Soc.* **159**, A1034–A1039 (2012) & *J. Phys. Chem. C* **119**, 4612–4619 (2015)]: $Z_{\omega} = \sqrt{[R_{ion,L} / (j\omega C_{dl,A} \cdot 2\pi r)] \cdot \coth \sqrt{(R_{ion,L} \cdot j\omega C_{dl,A} \cdot 2\pi r L)}}$. Fig. 3b was fitted in the simulation using this equation.

Fitting analysis results based on an equivalent circuit have been added to Supplementary Fig. 8.

Reviewer #3 (Remarks to the Author):

This paper reported a new designed ultra-light weight Li-S pouch cell, using Sulfurized polyacrylonitrile as cathode active material combining with ten state-of-art technologies, achieved the highest gravimetric energy density of 761Wh/kg, which is very promising for the next-generation batteries. The manuscript can be considered for publication. Below are my comments for the authors,

Answer: Thank you for your positive review for our study. Your comments were very helpful in the development of our study.

1) The research combines ten state-of-art technologies to achieve the goal, would the authors know which technologies are the most important factors? Have the authors considered or tried to fabricate a baseline cell and fabricate cells only with one technology to compare? It would be interesting to see the differences caused by single factor.

Answer: All state-of-art technologies are definitely important for the ultra-lightweight Li-SPAN pouch cell with >750 Wh/kg-cell, technologies I,III,VI,VII,VIII,X are particularly effective in achieving higher gravimetric energy density. We investigated with small capacity baseline cells to test the effect of each technology element and used the results as bases for the final Li-SPAN pouch cell design. However, the final cell design could not be achieved by simple addition of each technology elemental. It was necessary to successfully harmonize each technology with engineering and chemistry.

2) This paper only mentioned cycling tests using very low C rate under 30 degC, how about its performance in fast charging or low temperature?

Answer: Thank you for your comment. This is an important point. There is a trade-off between C-rate characteristics and energy density in this Li-SPAN pouch cell. The pouch cell can therefore be used below 0.2C-rate. The C-rate performance of the Li-SPAN pouch cell is relatively good, as lightweight Li-S cells to date have often been limited to 0.05C or less [*J. Energy Chem.* **76**, 181–186 (2023)]. The explanation was added in the revised manuscript. Low temperature performance will be evaluated in the future.

3) What would be the difficulties for the new cell towards mass production? What are the future plans from the authors?

Answer: Issues for mass production of the new battery cells in a practical phase include establishment of manufacturing processes and factory designs, mass productions of various materials and equipments, module/pack assembly, operation and safety tests, standards for use, and others. We

will contribute to practical use of the new battery cells by mass producing and improving SPAN powders and by designing and demonstrating Li-SPAN cells with higher battery performances.

Specifically,

4) Wonder why the author chose to do the cycling tests under 30 degC, not 25 degC, 40 degC or other temperature points?

Answer: Stability tests at 25 °C were difficult due to the experimental environment and equipment, so the tests were carried out at 30 °C. This means that the 30 °C tests were positioned as a standard test condition close to room temperature.

5) In Technology VII part, is 'cyclized-PAN' a typo? Should it be 'cyclized-SPAN'?

Answer: SPAN is considered to consist of sulfur components and cyclized-PAN backbones. The 'cyclized-PAN' is not a typo and indicates backbone parts excluding sulfur components.

6) In Technology VIII part, why the quality of the CEI film was poor in the case of the Light-Ele?

Answer: The CEI film analyses will be carried out in the future. It is inferred that the electrochemical decomposition products of the Light-Ele are more unstable than in the case of carbonate-based electrolyte solutions, making it difficult to form homogeneous and ionically conductive films.

7) In Ultra-lightweight cell part, from Supplementary Fig. 16 it showed different charge/discharge C rates were used, and the voltage window in cycling tests were also different, is it appropriate to get the conclusion 'The number of chg./dischg. cycles can be improved by reducing the cell energy density'?

Answer: At present, there is a trade-off between chg./dischg. cycle lifespan and energy density in the Li-SPAN pouch cells. The required optimal balance of cycle lifespan and energy density varies depending on commercial applications. We will demonstrate various Li-SPAN cell performances and expand the application possibilities. Furthermore, we will continue to challenge Li-SPAN battery designs with both longer cycle lifespan and higher gravimetric energy density.

Main text has been revised.

Supplementary Fig. 4 has been revised.

Supplementary Fig. 8 has been revised.

Supplementary Table 4 has been revised.

REVIEWERS' COMMENTS:

Reviewer #1 (Remarks to the Author):

The revised manuscript meets the publication standards of the journal.

Reviewer #2 (Remarks to the Author):

it is revised accordingl.

Reviewer #3 (Remarks to the Author):

Thanks for addressing my points, I am satisfied with the author's answers.